# The Structure of the Cardiac Mitochondria Respirasome Is Adapted for the β-Oxidation of Fatty Acids

**DOI:** 10.3390/ijms25042410

**Published:** 2024-02-18

**Authors:** Alexander V. Panov

**Affiliations:** Department of Biomedical Sciences, School of Medicine, Mercer University, Macon, GA 31201, USA; alexander.panov55@gmail.com

**Keywords:** heart mitochondria, β-oxidation of fatty acids, respiratory chain, respirasome, ubiquinone, ubiquinol, oxidative phosphorylation, tricarboxylic acid cycle

## Abstract

It is well known that in the heart and kidney mitochondria, more than 95% of ATP production is supported by the β-oxidation of long-chain fatty acids. However, the β-oxidation of fatty acids by mitochondria has been studied much less than the substrates formed during the catabolism of carbohydrates and amino acids. In the last few decades, several discoveries have been made that are directly related to fatty acid oxidation. In this review, we made an attempt to re-evaluate the β-oxidation of long-chain fatty acids from the perspectives of new discoveries. The single set of electron transporters of the cardiac mitochondrial respiratory chain is organized into three supercomplexes. Two of them contain complex I, a dimer of complex III, and two dimers of complex IV. The third, smaller supercomplex contains a dimer of complex III and two dimers of complex IV. We also considered other important discoveries. First, the enzymes of the β-oxidation of fatty acids are physically associated with the respirasome. Second, the β-oxidation of fatty acids creates the highest level of QH_2_ and reverses the flow of electrons from QH_2_ through complex II, reducing fumarate to succinate. Third, β-oxidation is greatly stimulated in the presence of succinate. We argue that the respirasome is uniquely adapted for the β-oxidation of fatty acids. The acyl-CoA dehydrogenase complex reduces the membrane’s pool of ubiquinone to QH_2_, which is instantly oxidized by the smaller supercomplex, generating a high energization of mitochondria and reversing the electron flow through complex II, which reverses the electron flow through complex I, increasing the NADH/NAD^+^ ratio in the matrix. The mitochondrial nicotinamide nucleotide transhydrogenase catalyzes a hydride (H^-^, a proton plus two electrons) transfer across the inner mitochondrial membrane, reducing the cytosolic pool of NADP(H), thus providing the heart with ATP for muscle contraction and energy and reducing equivalents for the housekeeping processes.

## 1. Introduction

In the middle of the 20th century, physiologists showed that in the heart and kidneys, β-oxidation of the long-chain fatty acids (FAs) provides more than 95% of energy for ATP production [1,2,3,4]. However, researchers studying respiratory activities of the isolated mitochondria relatively rarely used long-chain fatty acids as substrates. In comparison with other commonly used respiratory substrates, which are products of carbohydrates or amino acid catabolism, fatty acids are much more complicated as substrates for mitochondria. They are divided into short-chain (C2–C4), middle-chain (C6–C12), and long-chain (C14–C20) according to the length of the aliphatic chain, and they may have different numbers of double bonds. Different organs may have different propensities to the oxidation of fatty acids and different preferences to the length of the carbon chain. Polyunsaturated long-chain fatty acids have important biological roles and are not oxidized by the mitochondria. Unsurprisingly, there is great controversy regarding the oxidation of fatty acids and different opinions regarding their role in different organs as a source of energy.

The situation with mitochondrial respiration became even more complicated after it was discovered that the respiratory chain is organized into three supercomplexes, which together make up a more complex structure named the respirasome [5,6,7]. The physiological and functional significance of the respirasome could not be understood from our previous experimental information about the structure and function of the mitochondrial respiratory chain (Figure 1). For many decades, we did too many things to distort the truth. Here are several examples: (1) More than 80% of our knowledge about mitochondria was obtained by studying liver mitochondria cheaply and quickly. However, liver mitochondria from starved animals do not oxidize pyruvate, and researchers used either glutamate or, more often, succinate + rotenone instead of fatty acids. In addition, the liver has many unique functions and cannot serve as a standard for mitochondria from other organs. (2) In experiments with the isolated mitochondria, researchers practically always used a single substrate, sometimes with malate, whereas in vivo mitochondria oxidize substrates from several metabolic pathways simultaneously. (3) The energy metabolism of most organs was considered and studied from the perspective of the tricarboxylic acid cycle. Thus, brain mitochondria are believed to utilize only glucose or lactate as the main energy source [8]. However, it was established that even the isolated brain synaptic mitochondria, the isolated heart, and kidney mitochondria perfectly oxidize long-chain FAs in the presence of succinate, glutamate, or pyruvate [9,10,11,12,13]. (4) In experiments with the isolated mitochondria, many researchers assigned pyruvate and glutamate as substrates for complex I and succinate for complex II. However, glutamate dehydrogenase is located mostly in hepatocytes, whereas in other organs, the predominant oxidation of glutamate and often pyruvate occurs via the transamination pathway with the formation of succinate [14].

Of course, during previous decades, many great discoveries were made. However, mitochondrial and cellular physiologies were excluded from the in vitro experiments as a goal. Most importantly, the selection of substrates and often incubation conditions was far from the real conditions in the organs. In our publications, we have stressed that only long-chain fatty acids can maintain high rates of ATP production for a long period of time [10,11,12,13,15]. In this work, we propose that in the heart, the respirasome is evolutionarily adapted for the effective oxidation of long-chain and middle-chain fatty acids to maintain high rates of ATP production and also to stimulate anabolic and anaplerotic metabolic pathways in the hard-working organs, such as the heart, kidneys, brain, and skeletal muscles.

## 2. The Superstructural Organization of the Respiratory Chain

Back in the early 1980s, it was shown that a single set of respiratory complexes and ATP synthases for cardiac mitochondria has the following ratios: complexes I:II:III:IV:V related as 1:2:3:6–7:3–5 [16]. These ratios could not be understood in terms of existing models of the respiratory chain structure shown in Figure 1.

**Figure 1 ijms-25-02410-f001:**
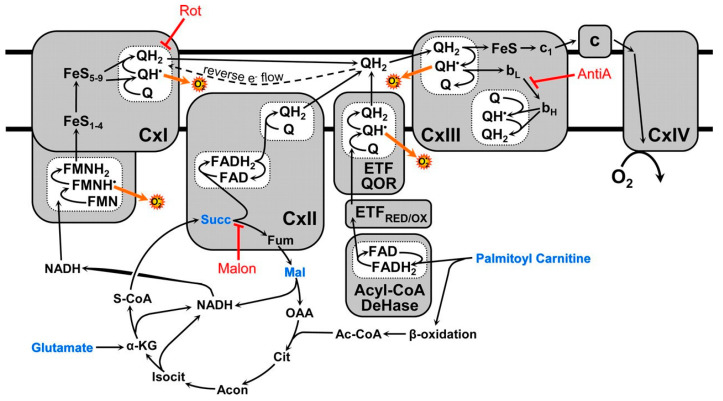
Schematic presentation of the mitochondrial respiratory chain and ATP synthase. Mitochondrial pathways of electron flow resulting from the substrates and inhibitors used in this study. The substrates used were glutamate/malate (which generates NADH via the tricarboxylic acid cycle, feeding into complex I), succinate (which feeds electrons directly into complex II), and palmitoyl-carnitine (which feeds electrons into the ETC via acyl-CoA dehydrogenase as well as through the β-oxidation pathway). The inhibitors used were rotenone, which inhibits complex I at the downstream Q binding site, malonate (a competitive inhibitor of complex II), and antimycin A (a complex III inhibitor that prevents electron flow to the QI site of complex III), thus stabilizing QH^*^ at the QO. The figure was adapted from [17].

Figure 1 and similar presentations of the respiratory chain indicate only the principal sequence of electron movement along the respiratory chain and are generally misleading. In 2000, it was shown that the single set of electron carriers is organized into three supercomplexes together, forming a functional unit named the respirasome [5,6]. The two large supercomplexes comprise one complex I associated with one dimer (two copies) of complex III connected with two dimers of complex IV (four copies). The third, smaller supercomplex contains one dimer of complex III associated with two dimers of complex IV (Figure 2).

Since Schagger’s publications about the organization of the respiratory chain into three main supercomplexes, many papers have been published regarding various aspects of the supercomplexes. From the literature, it becomes apparent that the term “respirasome’s superstructure” describes a phenomenon of formation of membrane-bound clusters of respiratory complexes rather than entities with a well-defined composition [18]. Dudkina et al. (2010) named these clusters “respiratory string”, as shown in Figure 3 [19].

Nesterov et al. (2022, 2023) developed a dynamic model of the long-range transport of energized protons along the mitochondrial inner membrane accompanied by the collective excitation of localized waves propagating on the membrane surface. This model is based on the new data on the macromolecular organization of the oxidative phosphorylation system (OXPHOS), showing the well-ordered structure of respirasomes and ATP synthases on the cristae membrane folds (Figure 4) [20,21].

An analysis of the state of respirasomes in patients with an isolated deficiency of single complexes suggests that forming respirasomes is important for the assembly/stability of complex I, the major entry point of respiratory chain substrates. Genetic alterations leading to a loss of complex III prevented respirasome formation, resulting in the secondary loss of complex I [22]. Only a few of the published papers specifically mentioned the physiological role of the respirasome. Still, they discussed electron transport from the perspective of glucose metabolism (pyruvate) and the Krebs cycle [23]. Some of the authors recognized that the physiological significance of the respirasome superstructure remains an enigma [24,25,26] but did not even mention how the respirasome might participate in the β-oxidation of long-chain fatty acids (LCFAs).

## 3. The Structural–Functional Properties of the Cardiac Mitochondrial Respirasome Evidence That the Respirasome Is Specifically Adapted for the β-Oxidation of Fatty Acids

All authors, however, who studied the properties of respiratory supercomplexes in different species recognize that these structures are highly dynamic and evidently can accommodate the particular metabolic demands of the species [18,25,26,27]. Accepting this point of view and the established structure of the heart mitochondria respirasome [5,6], let us think about the heart’s metabolic demands. We know the following: (1) The heart works constantly and consumes significant amounts of ATP, which requires long-lasting substrates; fatty acids are the only choice. (2) The heart works in a wide range of functional loads. (3) The heart’s cardiomyocytes, to a large degree, work as a syncytium [28]. (4) About 95% of the energy cardiomyocytes obtain is from the β-oxidation of long-chain fatty acids [1]. (5) Two enzyme complexes responsible for the β-oxidation of fatty acids are physically attached to the respirasome [29]. Since the electron-transporting complexes that constitute the respirasome’s supercomplexes are tightly packed [19,20,21], physical contact with the beta-oxidation enzymes will promote the membrane’s reduction/oxidation cycle of the ubiquinone pool.

It is evident that the smaller supercomplex, with the active centers of the complex III dimer open inside the inner mitochondrial membrane, uses the reduced coenzyme Q (ubiquinol) as the source of hydrogen. Oxidation of ubiquinol occurs at a very high rate. The two larger supercomplexes utilize acetyl-CoA produced by the trifunctional protein of the fatty acid oxidation system or by decarboxylation of the glycolytic pyruvate, as well as the substrates of the tricarboxylic cycle, as the source of hydrogen in the form of NADH + H^+^.

The reduction in the membrane pool of Co-Q to Co-QH_2_ occurs during the β-oxidation of FAs via the work of FAD-containing enzymes of the acyl-CoA dehydrogenase complex (Figure 5) and by succinate dehydrogenase, also known as Complex II. In the liver and the islet cells of the pancreas, glycerol-3-phosphate may be involved in reducing the membrane’s ubiquinone. However, this pathway of ubiquinone reduction does not play an essential role in the brain, heart, kidneys, and white fat tissue [30].

The highest rates of ubiquinone reduction to ubiquinol occur in the organs, where the β-oxidation of long-chain fatty acids is the main energy source. Correspondingly, in these organs, the highest steady-state levels of QH_2_ are maintained [31].

### 3.1. In the Absence of β-Oxidation of Long-Chain and Middle-Chain Fatty Acids, the Respirasome Predominantly Supports the Catabolic Reactions

Figure 6 presents a metabolic situation typically occurring during the in vitro experiments with the isolated mitochondria oxidizing any substrates but fatty acids. Without the β-oxidation of fatty acids, the TCA cycle enzyme succinate dehydrogenase (SDH) is mitochondria’s only ubiquinol (QH_2_) source. Because the smaller subunit of the respirasome lacks complex I, it directly interacts and instantly oxidizes the membrane’s ubiquinol. The extremely high rate of QH_2_ oxidation is directly associated with the structure of the smaller respirasome subunit, which has two active centers at the dimer of complex III, each of which reacts with a dimer of complex IV that catalyzes the irreversible reaction of water formation and releases a large portion of energy as heat. This is the point of irreversibility for mitochondrial energy metabolism.

However, in vivo, the resources of succinate are too small to become the main substrate for energization. Therefore, the maximal respiration rate is limited by the rate of succinate formation. Other mitochondrial metabolites, which are formed or metabolized after entering the tricarboxylic acid cycle (see Figure 6), provide electrons to the respiratory chain in the form of NADH + H^+^. It is well established that the NADH dehydrogenase of complex I is the rate-limiting step in mitochondrial respiration based on the NAD-dependent substrates [14]. In addition, in the experiments in vitro, it was established that, particularly in the mitochondria isolated from the brain or heart, the rate of externally added succinate oxidation is controlled by the phenomenon called “the intrinsic inhibition of SDH”. The inhibition is caused by endogenous oxaloacetate, which is discussed more thoroughly in [31,32,33]. Notably, kidney mitochondria are the only ones in the human body where the intrinsic inhibition of SDH is absent, and mitochondria can accumulate succinate during the β-oxidation of fatty acids [1,13].

Figure 7 presents respiratory rates of the isolated rat heart mitochondria oxidizing various substrates and their mixtures during resting (State 4), active ADP phosphorylation (State 3), and uncoupled respiration (State 3U).

Figure 7A–C show that “classical substrates” for complex I provide only moderate rates of oxidative phosphorylation in the rat heart mitochondria. However, the inhibitor analysis has shown that in the brain and particularly in the heart mitochondria, these substrates are oxidized via transamination with the formation of α-ketoglutarate, which is further oxidized to succinate [14]. Figure 7D,E show that the intrinsic inhibition of succinate oxidation was abolished in the presence of pyruvate or glutamate. Figure 7F shows that palmitoyl-carnitine alone is a very bad substrate for the heart mitochondria. This was the reason why researchers almost never utilized long-chain acyl-carnitines as substrates for the heart mitochondria.

However, Figure 8A shows that when the two “bad” substrates succinate (Figure 7D) and palmitoyl-carnitine (Figure 7F) are added together, the respiration rates increase dramatically in all metabolic states. The rates of ADP phosphorylation increased to the maximum for this type of mitochondria. The respiration rates were also high when succinate was mixed with pyruvate or glutamate. With regard to malate, we can mention that different animals, even from the same strain, respond differently upon the addition of malate: in some animals, malate increased the stimulatory effects of pyruvate or glutamate on succinate or palmitoyl-carnitine oxidation; in others, malate significantly inhibited these effects [35].

From the data presented in Figure 7 and Figure 8, we can suggest that in the absence of β-oxidation of the FAs, the large supercomplexes of the respirasome functionally are not designed to support the high rates of ATP consumption as is in vivo in the heart, kidneys, and brain [14]. Only long-chain fatty acids can sustain high rates of ATP production for a long time [34].

### 3.2. β-Oxidation of Long-Chain Fatty Acids in the Presence of Other Mitochondrial Substrates Supports a High Rate of ATP Production and Anabolic Metabolism in Cardiomyocytes

For the last 20 years, after discovering the respirasome’s structure, many researchers have studied various aspects of the respirasome structure and function [18,19,20,21,22,23,24,25,26,27]. Unfortunately, most of these studies were based on old paradigms. They did not address the physiological aspects of the respirasome in the oxidation of the body’s main substrates, long-chain fatty acids. Meanwhile, other researchers made great discoveries, which also initially met little attention from other researchers but directly contributed to understanding mitochondrial energy metabolism [5,29,31,34]. In 2010, it was shown that the enzymes of the β-oxidation of long-chain fatty acids are physically attached to the respirasome [29]. Brand and his colleagues have shown that the highest stationary levels of ubiquinol are maintained in the organs, where the β-oxidation of long-chain fatty acids is the main energy source [31,32,33]. Moreover, Brand and his team discovered that at a high level of ubiquinol, succinate dehydrogenase reverses the flow of electrons from ubiquinol into mitochondria and reduces fumarate to succinate [31,32,33].

Finally, it has been shown that β-oxidation of the long-chain fatty acids requires, for achieving maximum rates, the simultaneous presence of other mitochondrial metabolites: succinate, glutamate, or pyruvate [11,34].

Figure 9 illustrates the situation when the mitochondrial β-oxidation of LCFA is the main source of mitochondrial energization. According to Brand, during the β-oxidation of LCFA, mitochondria reduce the matrix pool of NAD to NADH + H^+^ and the membrane’s pool of ubiquinone to ubiquinol (QH_2_) [31,32,33]. It must be remembered that the enzymes involved in the β-oxidation of fatty acids are also arranged into two polyenzymatic complexes, which are physically associated with the respirasome [29]. The structure of the minor supercomplex of the respirasome allows ubiquinol to be oxidized extremely fast and thus maintains the highest demands in ATP. Electrons from the QH_2_ at the respiratory complex II (SDH) become reversed, thus turning the TCA cycle function from the catabolic metabolic pathway to anabolic and anaplerotic pathways. In well-energized mitochondria, the excess electrons reduce components of complex I and thus accelerate the production of superoxide radicals [34,35]. Under these conditions, the SDH also produces ROS at a high rate [31,32,33]. We have observed this in experiments with the isolated mitochondria [34] when incubation conditions did not allow mitochondria to activate nicotinamide nucleotide transhydrogenase (NNT). In situ, however, in the energized heart mitochondria, the activity of NTT will transfer the excessive energy into the cytoplasm by reducing NADP^+^ to NADPH, thus diminishing or even preventing the formation of superoxide radicals.

At high mitochondrial energization, the large supercomplexes of the respirasome maintain anaplerotic reactions, such as aerobic gluconeogenesis [9] and anabolic processes in the cytoplasm, which require NADPH. Energy-dependent mitochondrial nicotinamide nucleotide transhydrogenase maintains the cells’ high NADPH/NADP^+^ ratio [36,37,38]. The primary role attributed to the NNT’s forward reaction is maintaining an elevated cytosolic NADPH/NADP^+^ ratio. The cytosolic NADPH supply is critical to support various physiological functions, including biosynthetic pathways, mtDNA replication and maintenance, and enzymatic systems involved in thiol reduction and peroxide detoxification [39]. The highest expression of NNT was observed in the heart and kidney, which utilize the β-oxidation of fatty acids as the primary energy source [40].

Obviously, the activation of anaplerotic and synthetic reactions in the cell will depend on the characteristics of fatty acids metabolism and functions of the organ. For example, in brain astrocytes, the oxidation of fatty acids ensures the activity of aerobic glycolysis and the formation of lactate, which is a direct substrate for brain neurons. The anaplerotic formation of glutamine is a source of neurotransmitters glutamate and GABA [9,10]. All of these metabolic pathways are irreversible due to the smaller supercomplex of the respirasome.

## 4. Stimulation of β-Oxidation of Fatty Acids by Supporting Substrates

We studied the effects of mitochondrial metabolites, which are by themselves substrates for mitochondrial respiration, on the β-oxidation of palmitoyl-carnitine (a long-chain (C16) acyl-carnitine) using isolated mitochondria from three organs of a rat: heart, brain synapses, and mice kidney cortex as described in [11,34,35]. With palmitoyl-carnitine+ malate as the substrate, brain and heart mitochondria had very low rates of ADP phosphorylation. With succinate alone, brain and heart mitochondria showed no stimulation of the State 3 respiration upon the addition of ADP. However, when the brain and heart mitochondria were oxidizing palmitoyl-carnitine and succinate simultaneously, the rates of the State 3 respiration were highest and exceeded the State 3 respiratory rates for palmitoyl-carnitine + glutamate or palmitoyl-carnitine + pyruvate by 30–70% [34,35]. Remarkably, with the brain mitochondria, pyruvate was more effective in stimulating respiration with palmitoyl-carnitine than glutamate, whereas with the heart mitochondria, glutamate was more effective than pyruvate [34,35]. This coincides with the activities of glutamate-aspartate transaminase, which is higher in the heart than in the brain mitochondria.

Unlike the brain and heart mitochondria, kidney mitochondria lack the intrinsic inhibition of SDH and oxidized succinate at very high rates [11]. However, kidney mitochondria also showed low rates of respiration in all metabolic states with palmitoyl-carnitine + malate. The highest stimulation of palmitoyl-carnitine oxidation was observed with 5 mM succinate, whereas glutamate was much less effective than succinate, and pyruvate was completely ineffective. Because with the kidney cortex mitochondria, the State 3 succinate respiration rate was only 20–30% lower than with palmitoyl-carnitine + succinate, some of our colleagues doubted that succinate stimulated the oxidation of palmitoyl-carnitine. SDH in kidney mitochondria has a very high K_M_ for succinate; therefore, with 0.5 mM succinate, there was no respiration at all. However, 0.5 mM of succinate stimulated the rate of palmitoyl-carnitine oxidation 4-fold and that of octanoyl-carnitine 8-fold. These facts directly support the idea that succinate stimulates the β-oxidation of fatty acids. Moreover, the oxidation of fatty acids increases the concentration of ubiquinol and reverses the flow of electrons from QH_2_ to the TCA cycle, reducing fumarate to succinate [31]. Thus, the kidneys accumulate succinate and may play a major role in activating the succinate-specific G-protein GPR91 [13].

These data support our working hypothesis that the palmitoyl-carnitine oxidation is stimulated by succinate, and the stimulatory effects of pyruvate and glutamate depend on the transamination activities of the organ’s mitochondria. We suggest that succinate somehow, possibly by the allosteric mechanism, promotes the reversal of the electron flow from ubiquinol to fumarate and further to the large supercomplexes containing complex I. This working hypothesis agrees with our proposal that the β-oxidation of long-chain and middle-chain fatty acids is the main function of respirasome.

## 5. The Critical Roles of the Mitochondrial Phospholipids Cardiolipin and Phosphatidylethanolamine in Mitochondria’s Structural Organization and Functioning

The extensive work of many researchers allows us to appreciate a very complex organization of the mitochondrial oxidative phosphorylation system. Recent works on the mitochondrial respirasome, comprising three supercomplexes, suggest that the respiratory system for oxidation of a particular type of substrate has even higher orders of structural organization. In the case of the β-oxidation of fatty acids, the respiratory system includes the physical association of the respirasome with the enzymes of the β-oxidation of fatty acids [29] and succinate dehydrogenase of the TCA cycle (complex II) [40], which is part of the TCA cycle. In its turn, the respiratory functional megacomplex is structurally coupled with the ATP–synthase complex, forming a functional megastructure of an even higher order [20,21,41,42].

Previous ideas about the respiratory chain, as a sequence of electron carriers from NADH to oxygen, did not explain how the heart fulfills energy-consuming needs that are beyond contractile function. In this review, we argue that in a typical in vitro experiment, mitochondria oxidizing any substrate except fatty acids exhibit only catabolic properties and rarely exhibit maximum rates of ATP synthesis (see Figure 7 and Figure 8) [34]. High-energy-consuming organs, like the heart and kidneys, rely on fatty acid oxidation because this fuel source provides 106 ATP molecules compared to 36 from glucose metabolism [43]. Moreover, only the β-oxidation of fatty acids can provide the maximum rates of ubiquinol formation and, thus, maximum rates of respiration and ATP production, as well as support synthetic and anaplerotic functions for a long period of time.

The structure and functions of the oxidative phosphorylation system are inextricably linked with the unique mitochondrial phospholipids, phosphatidyl ethanolamine and cardiolipin, in particular [20,21,40,41,42]. In the mitochondrial membrane, cardiolipin (CL) is involved in the organization of multi-subunit oxidative phosphorylation complexes and their association with the higher-order supercomplexes [44]. Thus, not only the dysfunctions of proteins and the mutations of genes encoding them but also the oxidative or metabolic abnormalities of mitochondrial phospholipids may cause many diseases [40,41].

The ability of CL to fit into the negative curvatures of the inner membranes explains the fact that about 80% of CL is located in the inner leaf of the inner mitochondrial membrane, where CL interacts with a large number of mitochondrial proteins, complexes, and supercomplexes of the respiratory chain, ATP-synthase, ATP/ADP carrier, uncoupling protein, etc. [45]. The rest of the CL (about 20% of the total CL) is located in the outer leaf of the IMM, which, in general, has a positive curvature. But at the contacts of the outer leaf of the IMM, with the outer membrane, CL forms connecting complexes with negative curvature between the inner membrane and porin of the outer mitochondrial membranes, and also includes various intermembrane and cytosolic enzymes, for example, cytochrome *c*, hexokinase, creatine kinase, and ANT, which are enzymes specific for the metabolism of each organ [46]. These contact sites play an important role in the cristae organizing system and optimization of the organ’s energy metabolism [47]. Strong negative charges allow CL to interact with the membrane’s proteins and peptides by electrostatic interactions. Because the headgroup of CL is very small and bound to two phosphatides, the conformational variabilities are strongly limited [48]. This restricts intermolecular interactions of the head’s OH groups of cardiolipin in the rafts and with other phospholipids but makes the head’s phosphates open to interactions with the matrix water, metal ions, peptides, proteins, and lipids, which are much stronger as compared with other membrane lipids. Interactions of CL with other proteins may be so strong that Cl is found in the crystals of the isolated proteins, for example, in the crystals of Complex III and ANT [48]. Cardiolipin and phosphatidyl ethanol amine bind individual electron carriers into supercomplexes of the respirasome and then other enzymes into the higher-order structures.

The anionic properties of cardiolipin and its concentration at the negative curves of the cristae, which localize supercomplexes of the respiratory system, result in local acidification of the water layer close to the inner membrane. This increases the probability of the protonation of superoxide radicals produced by the respiratory supercomplexes and SDH [31]. The perhydroxyl radical (HO_2_^•^) specifically induces the isoprostane lipid peroxidation of polyunsaturated fatty acids that result in the oxidative damage of cardiolipin and phosphatidyl ethanol amine that participate in the formation and stabilization of the supercomplexes [49]. This is one of the most significant mechanisms of the organism’s overall aging and the cause of age-associated diseases [50].

## 6. Diseases Caused by Abnormalities of Fatty Acid Metabolism and Changes during Embryonic and Postembryonic Ontogenesis

Following from the hypothesis above, β-oxidation of the long-chain and middle-chain fatty acids is in direct structural and functional unity with other metabolic processes in the cell. The very high complexity of superstructures involved in fatty acid oxidation and associated anabolic processes leads to possible multiple pathways of damage and disturbances of the structure and functioning of this principal metabolic process. Here, we mention some of the pathologies caused by different mechanisms of disorders of fatty acid metabolism.

### 6.1. Cardiolipin Abnormalities

A mutation in the tafazzin (TAZ) gene causes Barth Syndrome, which is a rare systemic condition characterized by dilated cardiomyopathy, general weakness in the skeletal muscles, frequent infections due to neutropenia, and a lack of stamina. Barth syndrome occurs almost exclusively in males. Many affected children die of heart failure or infection in infancy or early childhood. Those who live into adulthood can survive into their late-forties. Tafazzin is the mitochondrial cardiolipin acyltransferase responsible for the “remodeling of cardiolipin” after its synthesis [51]. In Barth Syndrome, cardiac CL was composed of a random conformation instead of the usual symmetric linoleic-acid-rich form [52]. After the synthesis of CL de novo in the inner mitochondrial membrane, all four fatty acids are saturated and have random lengths. The unification of fatty acids in CL results from maturation. Most mammal animals in the matured CL have linoleic acids (C18:2) [53]. Unsaturated fatty acids in cardiolipin make it susceptible to oxidative damage [54]. In humans, all four inner fatty acids are represented by linoleic acid; however, during aging and diabetes, the fatty acids of the heart mitochondria cardiolipin often become replaced by arachidonic acid (C20:4) and docosahexaenoic acid (C22:6) [55,56]. In Barth Syndrome patients, the presence of “immature” CL results in structural changes in the respiratory chain supercomplexes and disturbances of many other normal cardiolipin functions. Dudek et al. (2016) observed a cardiac-specific loss of succinate dehydrogenase (SDH), which links the respirasome with the tricarboxylic acid cycle and thus breaks the anabolic and anaplerotic events supported by fatty acid oxidation [53]. However, as mentioned above, the linoleic acids of CL relatively easily undergo isoprostane lipid peroxidation. Therefore, the appearance of oxidized CL is a reliable marker for the aging of mitochondria [57]. As a consequence, numerous nonspecific damages accumulate and, in some individuals, may result in the development of the so-called “frailty syndrome” and “chronic fatigue syndrome” [58].

### 6.2. Enzyme Deficiencies

Metabolic cardiomyopathies can be caused by disturbances in fatty acid uptake and transport and metabolism, for example, diabetes mellitus, hypertrophy, heart failure, or alcoholic cardiomyopathy [59,60]. Fatty acid oxidation (FAO) disorders may be caused by mutations in more than 10 genes coding for the various enzymes and transporters involved in the pathway. As a group, FAO disorders belong to the most prevalent monogenic conditions worldwide [61]. Aberrations of nuclear and mitochondrial DNA lead to a wide variety of cardiac pathologies. A deficiency in the enzymes of the mitochondrial β-oxidation shows various cardiac dysfunctions [62]. Carnitine deficiency can be caused by both genetic and environmental causes, with resultant signs and symptoms of metabolic disease, including cardiomyopathy. The leading cause of all pathologies is the inefficiency of the fatty acid β-oxidation [63]. In the human heart, metabolic genes exist in constitutive and inducible forms. The failing adult heart reverts to a fetal metabolic gene profile by downregulating adult gene transcripts rather than by upregulating fetal genes [64]. In this respect, it is useful to recollect some features of the heart’s fetal and postnatal maturation of mitochondrial energy metabolism. 

### 6.3. Postnatal Maturation of the Mitochondrial Fatty Acid β-Oxidation

During the fetal development of the heart, energy is produced within the cardiac muscle cells, essentially relying on carbohydrates [65,66]. The major reason for this is a lack of mitochondrial oxidative phosphorylation, which requires postnatal maturation. Glycolysis is particularly important because it provides branchpoint metabolites for several biosynthetic pathways that are essential for the synthesis of nucleotides and nucleotide sugars, amino acids, and glycerophospholipids that support the proliferation of cardiomyocytes [67]. Later, closer to the birth and several weeks after birth, lactate and ketone bodies become important substrates for the production of ATP [68,69]. The maturation of mitochondrial oxidative phosphorylation, which comprises intracellular architecture and mitochondrial enzymes, requires time. In experimental animals, the “mature” β-oxidation of long-chain fatty acids begins 3 months after birth [70].

## 7. Conclusions 

The hypothesis presented in this review is a typical example of serendipity, or “connecting points”, each of which is important but neglected scientific discovery. As a result, we have a new picture of how heart mitochondria oxidize long-chain fatty acids and what physiological consequences result from this process. The respirasome, which comprises three separate supercomplexes, has physical contact with the two enzyme complexes of the β-oxidation of fatty acids: the acetyl-CoA dehydrogenase and the trifunctional protein. The smaller respirasome’s supercomplex instantly oxidizes CoQ-H_2_, creating a high transmembrane potential (Δp = ΔΨ − Z ΔpH), whereas the two large supercomplexes containing complex I transfer the energy into the cytoplasm of the cardiomyocytes in the form of a high NADPH/NADP^+^ ratio. Simultaneously, at a high speed, mitochondria generate large amounts of ATP. The high steady-state level of ubiquinol in the membrane reverses the electron flow in complex II (SDH), reducing fumarate to succinate. This redirects the metabolites of the tricarboxylic cycle to anaplerotic synthetic metabolic pathways. Most organs in the body possess the mitochondrial dicarboxylate carrier that mediates the electroneutral export of succinate across the mitochondrial inner membrane [71]. Thus, upon accumulation of succinate in the mitochondria, the cytosolic and nucleus pools of succinate become equilibrated. GPR91, also known as succinate receptor 1 (SUCNR1), faces the extracellular environment and responds to succinate with a half-maximum effective concentration of 28–56 μM. The highest succinate concentration reported for extracellular fluids was 200 µM [71]. GPR91 and its succinate ligand function as detectors of local stress, including ischemia, hypoxia, toxicity, and hyperglycemia. Local levels of succinate in the kidney activate the renin–angiotensin system, thus contributing, through GPR91, to developing hypertension and the complications of diabetes mellitus, metabolic disease, and liver damage [72]. Since the rate of the fatty acid β-oxidation depends on supporting substrates (succinate, glutamate, pyruvate) [12], this may explain the gender differences in physical performance and the heart’s accelerated aging in men [73,74].

## Figures and Tables

**Figure 2 ijms-25-02410-f002:**
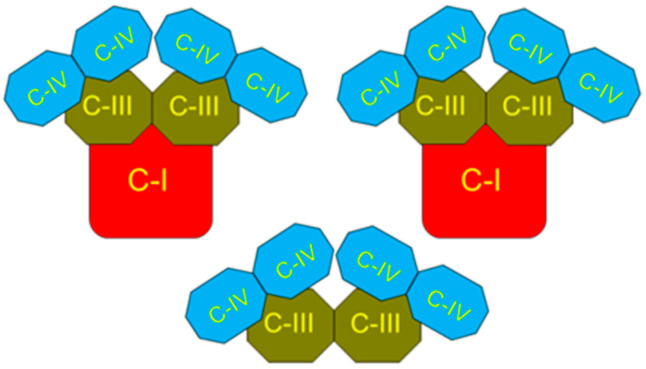
Schematic presentation of the respirasome. View from the matrix side on the two large supercomplexes and one smaller supercomplex. Complexes I, III, and IV are integral proteins. They penetrate the inner membrane and work as proton pumps. The figure is based on the data presented in [5]. Figure 6 and Figure 9 show more clearly how the respirasome’s supercomplexes might integrate into the inner membrane of mitochondria.

**Figure 3 ijms-25-02410-f003:**
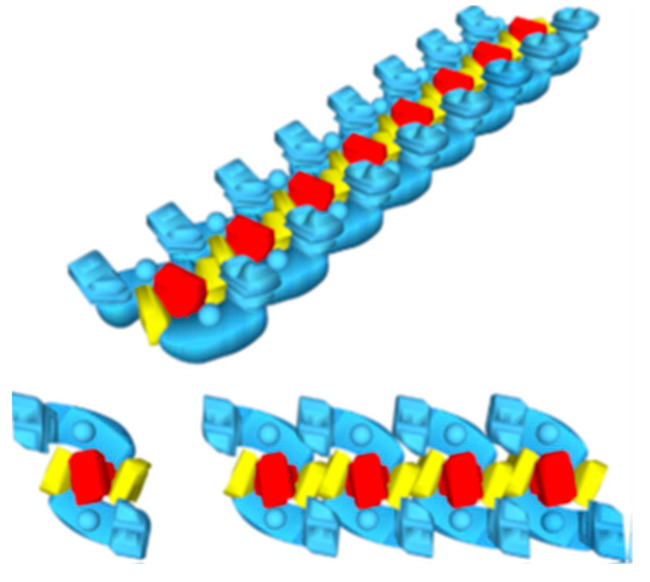
A schematic model of organizing respiratory chain supercomplexes into a respirasome and then to the respiratory string. The basic unit (lower left) consists of two copies of complex I (blue), one copy of complex III2 (red), and two copies of complex IV (yellow). The figure was adapted from [19].

**Figure 4 ijms-25-02410-f004:**
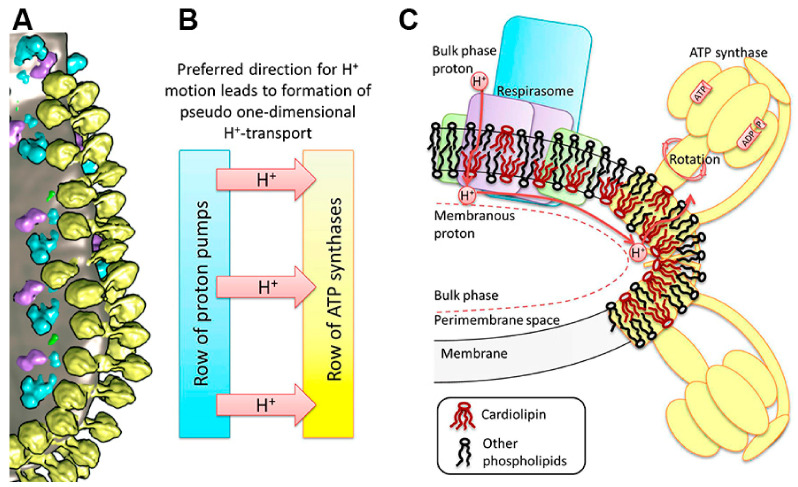
Structure of the mitochondrial OXPHOS system and cristae membrane illustrating a proton transfer pathway. (**A**) The cluster of components of the OXPHOS system at the bends of the cristae of heart mitochondria. Yellow—ATP synthase dimers; blue—complex I; purple—complex III dimers; green—complex IV; and grey—lipid membrane. (**B**) A dedicated direction of proton transfer between rows of proton pumps and ATP synthases. (**C**) Schematic reconstruction of the cluster in the OXPHOS system on the membrane fold and a pathway of the lateral transfer of protons from the respirasome to ATP synthase. The area of increased curvature of the membrane is enriched with CL molecules. The figure was adapted from [21].

**Figure 5 ijms-25-02410-f005:**
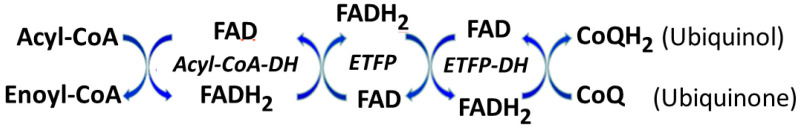
The sequence of reactions of formation of the trans-double bond between C-2 and C-3 thioesters of fatty acids and reduction of ubiquinone during the work of acyl-CoA dehydrogenase complex.

**Figure 6 ijms-25-02410-f006:**
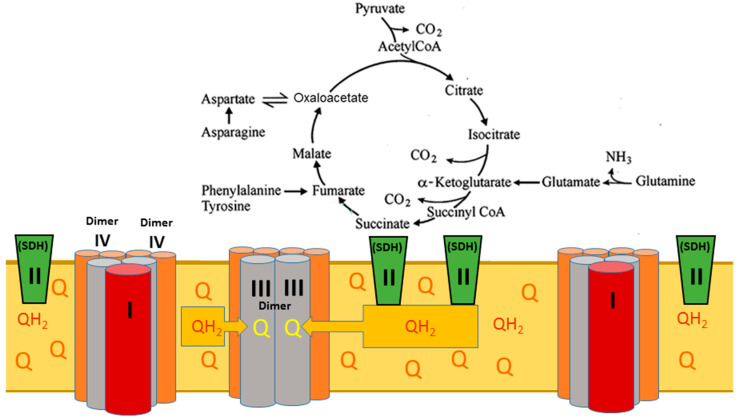
Without β-oxidation of fatty acids, succinate dehydrogenase is the only source of ubiquinol, and the mitochondrial metabolism becomes predominantly catabolic. Designations: Q—ubiquinone, the oxidized form of coenzyme Q; QH_2_—ubiquinol, the reduced form of coenzyme Q. The figure was adapted from [13].

**Figure 7 ijms-25-02410-f007:**
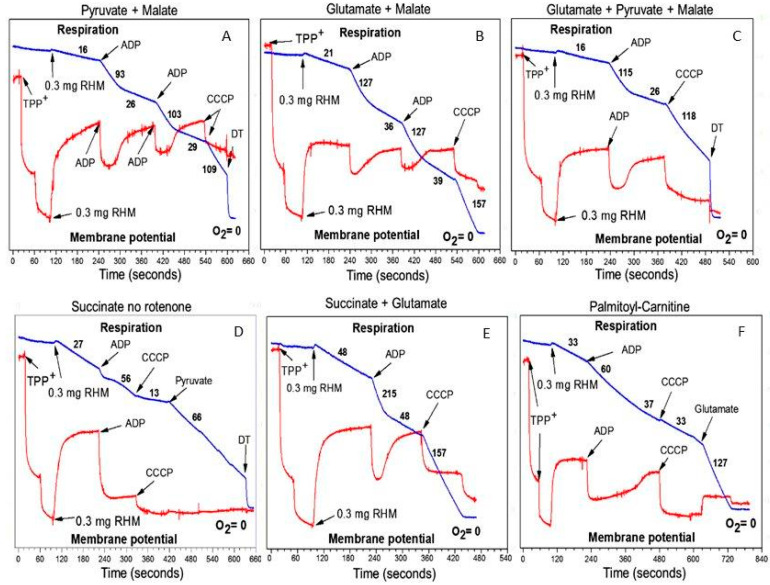
Oxidation by rat heart mitochondria of major substrates: (**A**) pyruvate 2.5 mM+ malate 2 mM; (**B**) glutamate 5 mM + malate; (**C**) glutamate + pyruvate + malate; (**D**) succinate 5 mM, no rotenone; (**E**) succinate + glutamate; (**F**) palmitoyl-carnitine 25 µM. Incubation conditions and experimental details are described in [14]. Additions: ADP 150 µM, CCCP 0.5 µM, rat heart mitochondria 0.3 mg, provide only (dithiothreitol) 10 µL of saturated solution. The figure was adapted from [34].

**Figure 8 ijms-25-02410-f008:**
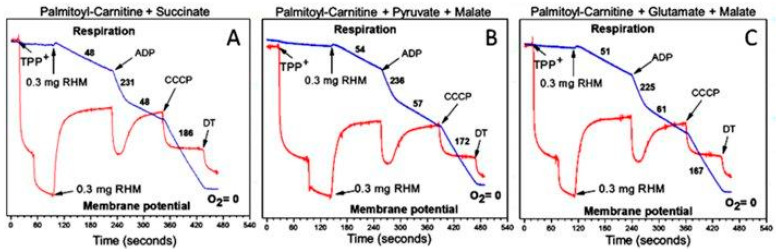
Oxidation by rat heart mitochondria palmitoyl-carnitine in the presence of supporting substrates: (**A**) palmitoyl-carnitine 25 µM + succinate 5 mM; (**B**) palmitoyl-carnitine 25 µM + pyruvate 2.5 mM + malate 2 mM; (**C**) palmitoyl-carnitine 25 µM + glutamate 5 mM + malate 2 mM. The figure was adapted from [34].

**Figure 9 ijms-25-02410-f009:**
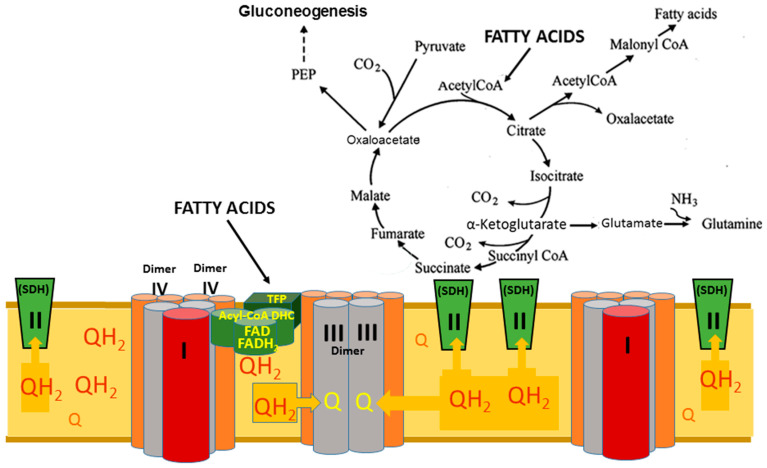
Functioning of respirasome and the Krebs Cycle during active β-oxidation of long-chain fatty acids. Abbreviations: Acyl-CoA DHC—acyl-CoA dehydrogenase complex, which includes three enzymes: acyl-CoA dehydrogenase, electron transfer flavoprotein (ETF), electron-transferring-flavoprotein dehydrogenase (ETFDH); PEP—phosphoenolpyruvate; TFP—trifunctional protein of the β-oxidation of fatty acids system; SDH—succinate dehydrogenase; Q—ubiquinone, oxidized form of coenzyme Q; QH_2_—ubiquinol, reduced form of coenzyme Q. The figure adapted from [13].

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
