# Peer review of "The Structure of the Cardiac Mitochondria Respirasome Is Adapted for the β-Oxidation of Fatty Acids"

_ijms, 2024, doi:10.3390/ijms25042410_

Round 1
Reviewer 1 Report
Comments and Suggestions for Authors
The review of Alexandr Panov “The structure of the cardiac mitochondria respirasome is adapted for the β-oxidation of fatty acids” describes the proposal that the respirasome is adapted to efficiently oxidize long- and medium-chain fatty acids to maintain high rates of ATP production in cardiac mitochondria.
There is no doubt that this review should be published in the journal IJMS, but the manuscript needs to be finalized.
1) The abstract must state the purpose of the review.
2) Introduction… Define paragraphs and change line spacing to suit the format of the entire text.
3) Line 47-48…”We must admit that almost everything we know about the structure and functions of respiration is incorrect.” A very bold statement by the author. However, this is a review and the author has the right to make similar statements.
4) Line 51-52…”It is not surprising that many scientists regard liver mitochondria as the standard for what mitochondria from other organs should be.”
I don't agree with this statement. I think that this sentence should be written more softly so as not to offend anyone. Not all scientists make liver mitochondria as a standard.
5) Line 146… FAs – to decipher where it is mentioned for the first time
6) Improve the quality of Figures 7, 8, and 9, if possible.
7) Lines 274 and 276….l.c. FAs or lcFAs - choose a single spelling
8) Line 287… NNT - decipher the abbreviation where it is mentioned for the first time
9) Line 287… “In situ…” replace with “In situ…”
10) Line 296… NADPH/NADP+ replace with NADPH/NADP+
11) Line 331… Km replace with Km
12) Line 383… сytochrome «c» replace with сytochrome c
13) In the conclusions, the author provides information about the inextricable connection between oxidative phosphorylation of phospholipids, in particular with cardiolipin. I agree that cardiolipin is a very important participant in the processes that the author describes. But these are conclusions, and the review does not contain any information about cardiolipin. I think it is necessary to describe the relationship between oxidative phosphorylation and cardiolipin.
That part of the conclusions in which cardiolipin is described should be moved to a separate chapter.
Comments on the Quality of English LanguageMinor editing of English language required
Author Response
The full response to the Reviewer 1 is the file attached below.

Reviewer 2 Report
Comments and Suggestions for Authors
In this review the author has discussed the organization of the mitochondrial respirasome in the context of heart and it’s energy requirements. The review is dense with information and makes the point that cardiac mitochondria is primed for performing beta fatty acid oxidation metabolism.
While reading through the review I had some concerns which are listed as follows:
· Line 287, 288 and 294
- please update the NNT vs NTT (they have been typos I believe.)
-also the abbreviation should appear the very first time and not on the third time.
· Line 274- I think the author means long chain Fatty acids where it is written l.c.FAs? please write in full or provide abbreviation list
· Line 255-257- the author writes “other researchers” but no reference has been provided – please include
Line 182- what does the author mean by ‘stationary’ QH2? Please elaborate
· Line 369- what is CL? Please write the full form.
· Line 58- please include references
· Line 364-366- please include references
· Line 80- the author discusses about issue of self-citation and in this review also 11 out of the 50 references cited are from the authors own work. I would highly encourage to acknowledge work of several other labs which have been and still are investigating other metabolic substrates in cardiac mitochondria a list of which can be found in another review PMID: 26474213 or in the following articles PMID: 10199888, PMID: 9823010; PMID: 1288863
· The author should mention in the review about the following aspects of cardiac metabolism
(a) the fact that metabolic shift occurs when the heart is suffering from a disease - PMID: 37533404; PMID: 26697888; PMID: 31842377; PMID: 11739307
(b) metabolic shift occurring while the heart develops from neonatal to an adult/mature stage- PMID: 30072919; PMID: 36220812; PMID: 10199888; PMID: 9823010; PMID: 1288863; PMID: 21209089
Comments on the Quality of English Language
Line 267- the first word is not written in English language. Please update
Line 28- does the author mean 60s ‘till’ the 20th century? It is not clear what the author means to convey by writing - 60s ‘of’ the 20th century . Please update/revise
Author Response
The full response to the Reviewer 2 is in the file attached below.
